# Semiconducting Metal Oxides: SrTiO₃, BaTiO₃ and BaSrTiO₃ in Gas-Sensing Applications: A Review

**Bartłomiej Szafraniak [1], Łukasz Fuśnik [1],*[ID], Jie Xu [2], Feng Gao [2], Andrzej Brudnik [1] and Artur Rydosz [1][ID]**

[1] Institute of Electronics, AGH University of Science and Technology, Al. Mickiewicza 30, 30-059 Kraków, Poland; szafrani@agh.edu.pl (B.S.); brudnik@agh.edu.pl (A.B.); rydosz@agh.edu.pl (A.R.)

[2] State Key Laboratory of Solidification Processing, NPU-QMUL Joint Research Institute of Advanced Materials and Structures (JRI-AMAS), School of Materials Science and Engineering, Northwestern Polytechnical University, Xi'an 710072, China; xujie@nwpu.edu.cn (J.X.); gaofeng@nwpu.edu.cn (F.G.)

\* Correspondence: lfusnik@agh.edu.pl; Tel.: +48-126-172-900

**Abstract:** In this work, a broad overview in the field of strontium titanate (ST, SrTiO₃)-, barium titanate (BT, BaTiO₃)- and barium strontium titanate (BST, BaSrTiO₃)-based gas sensors is presented and discussed. The above-mentioned materials are characterized by a perovskite structure with long-term stability and therefore are very promising materials for commercial gas-sensing applications. Within the last 20 years, the number of papers where ST, BT and BST materials were tested as gas-sensitive materials has ten times increased and therefore an actual review about them in this field has been expected by readers, who are researchers involved in gas-sensing applications and novel materials investigations, as well as industry research and development center members, who are constantly searching for gas-sensing materials exhibiting high 3S parameters (sensitivity, selectivity and stability) that can be adapted for commercial realizations. Finally, the NO₂-sensing characteristics of the BST-based gas sensors deposited by the authors with the utilization of magnetron sputtering technology are presented.

**Keywords:** strontium titanate (SrTiO₃); barium titanate (BaTiO₃); barium strontium titanate (BaSrTiO₃); gas-sensing applications; novel materials; nitrogen dioxide (NO₂) detection



## 1. Introduction

The first gas-sensing properties of semiconductor materials were reported in the 1920s, and in 2020, we celebrated the 100th anniversary of the first gas-sensing investigations of the gas atmosphere [1]. Since that time, semiconductor-based gas sensors reached a number of pivot moments including in 1955, when oxygen detection in gas changes in the conductivity of a semiconductor (ZnO) was reported [2]; in 1968, when the first commercially available gas sensor, TGS (Taguchi Gas Sensor from Figaro Engineering Inc., Arlington Heights, IL, USA), was launched on the market for methane detection TGS109 [3]; in the 1990s, when nanostructure gas sensors were presented, with zero-dimensional (0D) nanoparticles, one-dimensional (1D) nanorods and nanowires, two-dimensional (2D) nanosheets and films and three-dimensional (3D) polycrystals and ultraporous nanostructures as recently reviewed in [4]; and in 2007, when graphene-based gas sensors were introduced [5].

The common gases detected and analyzed by the commercially available gas sensors are oxygen, carbon monoxide, carbon dioxide, nitrogen oxides, ammonia, methane, acetone, methyl mercaptan, hydrogen sulfide, chlorine, volatile organic compounds (VOC) and hydrocarbon [6–8]. Nowadays, gas sensors are used in many applications [9–11], including mining [11,12], broadly understood industry [10,11], public safety [9–11], military [10], healthcare [10,11], consumer electronics [9,11,13], petrochemical [10,14], agriculture [10,11], water [11,14], medical [11,15–17], oil and gas [11,14], automotive [9,18,19], food and beverages [20], environmental protection [9,11,19], metals and chemicals [21,22], power stations [11], smart cities and building automation [10,11].

In the coming years, factors such as the growing demand for sensors in almost every field, technological progress, expansion of advanced functions of existing installations and devices, government requirements limiting emissions of various gases by industry and transport will accelerate the demand for this type of equipment (Figure 1). The value of the global gas sensor market was estimated at USD 2.19 billion in 2019. According to some sources [6,7], this market is forecast to grow by a compound annual growth rate (CAGR) of 6.3% to 8.3% from 2020 to 2027. One of the rapidly growing industries with an increasing demand for gas sensors is environmental monitoring, especially the level of pollution and volatile organic compounds.

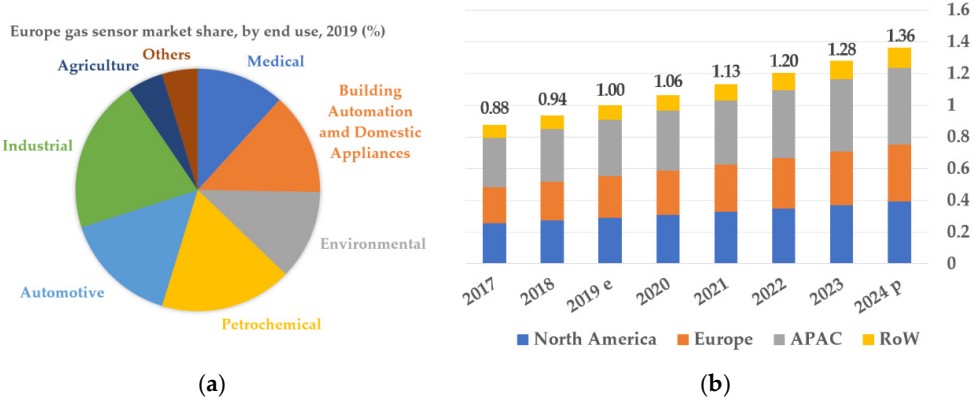

(a)
(b)

**Figure 1.** Selected statistical data of the global gas sensor market: (**a**) share of major sectors of the economy in the gas detector market for Europe; (**b**) global market for gas sensors broken down into regions—prediction; APAC—Asia Pacific, RoW—rest of the world, 2019 e—estimation, 2024 p—prediction [7,8].

Various materials have been validated for gas-sensing applications; however, the main commonality is gas sensors based on metal oxides, for example, n-type including tin dioxide ($SnO_2$), tungsten trioxide ($WO_3$), indium oxide ($In_2O_3$), gallium oxide ($Ga_2O_3$), vanadium oxide ($V_2O_5$) and iron oxide ($Fe_2O_3$) and p-type metal oxides such as nickel oxide (NiO), copper oxide (CuO), cobalt oxide ($Co_3O_4$), manganese oxide ($Mn_3O_4$) and chromium oxide ($Cr_2O_3$) [23]. The progress of nanotechnology development allows researchers to increase the surface-to-volume ratio by utilization of various techniques which results in increased sensitivity and a reduced operating temperature, and various doping methods and materials have been proposed to increase the selectivity and stability of gas sensors. Long-term stability is a crucial feature of gas sensors dedicated to industrial applications such as automotive, biotechnology, safety and military.

One of the promising materials is strontium titanate ($SrTiO_3$). $SrTiO_3$ is a semiconducting ceramic material and it has a simple cubic perovskite structure (space group *Pm3m*) with a lattice parameter of 0.3905 nm and a density of $r = 5.12$ g/cm$^3$ (Figure 2a) [24]. The crystallographic structure is presented in Figure 2. $SrTiO_3$ is attractive due to its properties such as a large dielectric constant ($\varepsilon_0 = 300$), low dielectric loss (mostly < 0.02) [25] and strong thermal and chemical stability [26]. These advantages enable a wide application of the material in the area of sensors, actuators, electro-optical devices, memory devices with random access and multilayer capacitors [27]. It is commonly used for oxygen sensors [28]; however, other sensing applications are also common such as temperature sensors [29] and the cantilever base for various sensors [30]. Thanks to the deposition technology developments, strontium titanate can be realized in the nanoform instead of the bulky form, where a high-temperature solid-state reaction method is used (700–1000 °C). Reducing the size allows researchers to reduce the operating temperature to 40 °C [31]. Among various possibilities to deposit $SrTiO_3$ such as magnetron sputtering [32,33], atomic layer deposition (ALD) [34,35], pulsed laser deposition (PLD) [36,37], metal-organic chemical vapor deposition (MOCVD) [38,39], laser chemical vapor deposition (LCVD) [40,41] and

the sol–gel method [42,43], the sol-gel method seems to be the most suitable since it enables depositing a small grain size with high uniformity and high purity. Nanoforms can be also obtained thanks to the glancing angle deposition technique, for example, with magnetron sputtering technology [44,45].

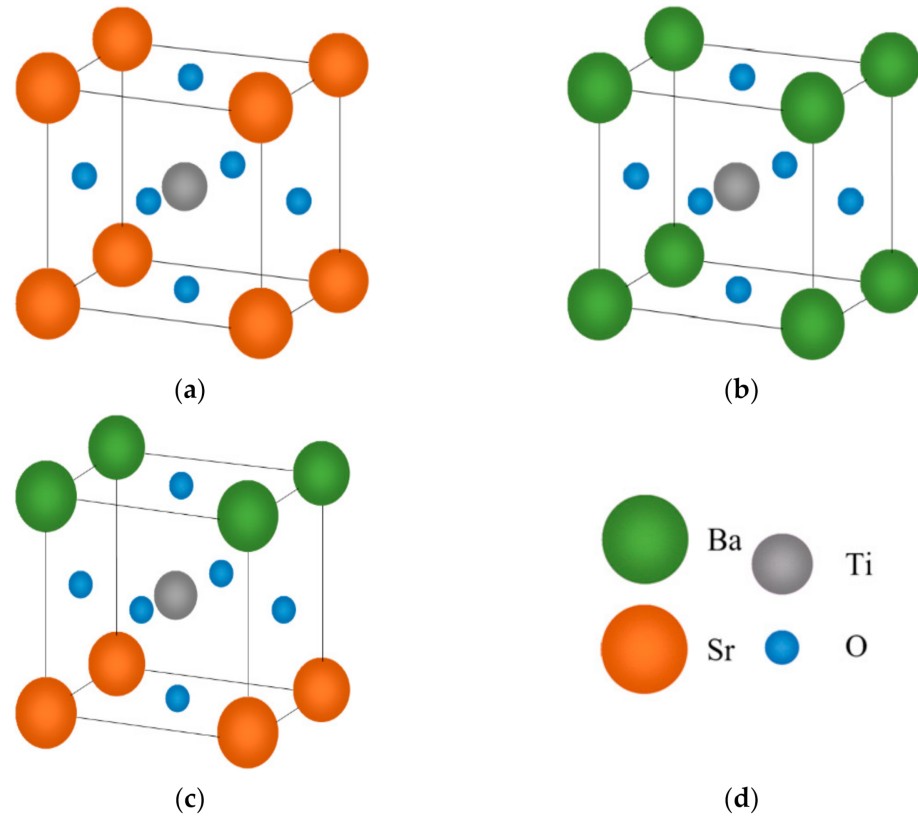

**Figure 2.** Non-polar structure at room temperature: (**a**) $SrTiO_3$; (**b**) $BaTiO_3$; (**c**) $BaSrTiO_3$; (**d**) legend.

Another promising material is barium titanate ($BaTiO_3$). $BaTiO_3$ is a cubic perovskite-type structure semiconductor material (Figure 2b), commonly used as a ferroelectric [46] since it exhibits a high dielectric constant (dielectric constant depends on the type of synthesis, temperature, frequency and dopants) [47,48], large electro-optic coefficients and a positive coefficient of resistivity (PTCR) [49]. There are various methods of obtaining $BaTiO_3$, for example, by a solid-state reaction [50], the sol–gel method [51], a hydrothermal method [52], a coprecipitation method [53], a polymeric precursor method [54] and mechanochemical synthesis [55]. Thanks to the above-mentioned features, BST is widely used in ferroelectric memories [56], electro-optical devices [57], dielectric capacitors [58], multilayer capacitors (MLCs) [59] and electromechanical transducers [60,61], as well as in gas sensor applications [62].

A novel material that brings all features of $SrTiO_3$ and $BaTiO_3$ is $BaSrTiO_3$ (barium strontium titanate, BST). BST is a kind of electronic ceramic material with a typical perovskite structure (Figure 2c) and properties such as a high dielectric constant, low dielectric loss, good tenability and high insulation resistance and can be widely used in various electronic components such as ferroelectric memories, capacitors and phase shifters [63–66]. Apart from the typical applications where the above-mentioned properties are used, BST has also been investigated as a gas-sensitive material.

In 2014, Romh MA E., et al. presented investigation results on the process of preparing and testing solid-state samples by mixing BT and BST powders with an organic carrier in the ratio of 50:50 and 60:40 and then depositing by spin coating on alumina substrates as a gas-sensitive material. $BaTiO_3$ was doped with strontium and iron to increase the conductivity by double substitution in the perovskite structure. The films were then sintered at the

temperature of 1100 °C for 2 h and characterized by means of X-ray diffraction (XRD) and a scanning electron microscope (SEM). The dielectric measurements performed by the authors revealed a significant increase in conductivity at a low a.c. current frequency, even 10,000 times at a temperature from 25 to 500 °C, and the phenomenon of dielectric relaxation related to the displacement of oxygen voids appeared [67].

In another work, Simion, C.E., et al. presented gas sensors based on thick-film BST doped with copper in various concentrations (0.1, 1 and 5 mol% Cu). The gas-sensing behavior was tested under exposure to $NH_3$ and $H_2S$. The prepared samples worked optimally at a temperature of 200 °C. The sample made of BST doped with 0.1% mol of Cu had the best sensitivity to $NH_3$ gas. On the other hand, selective detection of $H_2S$ was achieved for BST doped with 5 mol% Cu. The tested materials did not show the cross-sensitivity effect to CO, $NO_2$, $CH_4$ and $SO_2$ (200 °C, 50% RH). Higher relative humidity noticeably increased the sensitivity of the sensors to $NH_3$ and $H_2S$ [68]. Simion, Cristian E., et al. in the following year published another paper where they described the reactions of thick layers of $Ba_{0.75}Sr_{0.25}TiO_3$ to the presence of $NH_3$ at room temperature, at different levels of relative air humidity. The samples were synthesized by a hydrothermal method. The change in electrical resistance and capacitance was tested and photoacoustic measurements were conducted as well. The measurement results were presented in the context of the Grotthuss mechanism in relation to the ion/electronic conductivity in BST. The detection of ammonia was tested in the presence of water vapor at room temperature. As a result of the performed measurements, it was determined that the optimal detection of ammonia occurs at room temperature in the presence of water vapor in the tested gas. The interaction between $NH_3$ and $H_2O$ takes place mainly through the proton exchange mechanism [69].

In 2020, Shastri, Nipa M., et al. presented BST-based gas sensors obtained with the pulsed laser deposition (PLD) technique. The paper presents two stoichiometries of $Ba_xSr_{1-x}TiO_3$ with x = 0.5 and x = 0.7. The material properties were checked using the following methods: XRD, X-ray energy dispersion (EDX) and UV–VIS spectroscopy. The gas-sensing measurements showed that the $BST_{0.5}$ sample exhibited a higher sensitivity to $H_2S$ with a concentration of 800 ppm than the $BST_{0.7}$ sample [70]. Recently, $Ba_{0.5}Sr_{0.5}TiO_3$ doped with various concentrations of $RuO_2$ (from 0% to 6%) was used as a gas-sensing material with the utilization of chemical solution deposition (CSD) on p-type silicon substrates. The sample with the highest $RuO_2$ content (6%) exhibited the highest response to $H_2S$, which is considered as a biomarker of halitosis disease, and therefore the developed sensors ($Ba_{0.5}Sr_{0.5}TiO_3$ doped with $RuO_2$) were proposed for exhaled breath analysis [71].

In this paper, the gas-sensing achievements of $SrTiO_3$- (Section 2.1), $BaTiO_3$- (Section 2.2) and $BaSrTiO_3$-based gas sensors (Section 2.3) are summarized, giving the readers a frank overview. Table A4 in Appendix A presents a collective comparison of the properties of the above-mentioned three nanocomposites, it is a short and concise comparative characteristic. There is a literature reference next to each piece of information. As can be noticed in Figure 3, the number of papers where the above-mentioned materials were used for gas-sensing materials constantly increases from 2000.

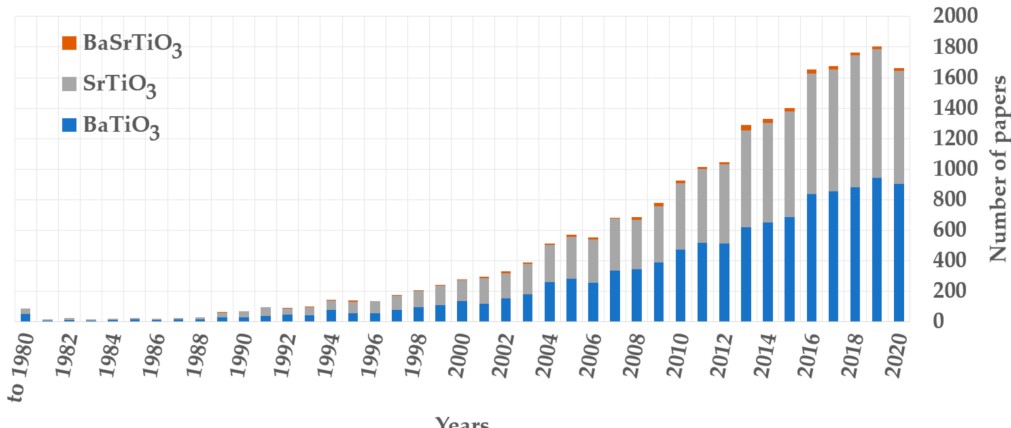

**Figure 3.** The number of papers related to SrTiO₃-, BaTiO₃- and BaSrTiO₃-based gas sensors from 1980 to 2020.

## 2. Materials, Results and Discussion

### 2.1. SrTiO₃ for Gas-Sensing Applications

In 2004, Meyer and Waser presented a model for a fast sensor response of resistive donor-doped SrTiO₃ at temperatures from 850 to 950 °C. The authors speculated that cation vacancies may play a key role in the formation of grain resistance boundaries not only at high but also at moderate temperatures. As a result of mobility, the change in cationic vacancy concentration may be limited to only a few monolayers on each side of the interface. Strontium vacancies are considered to be virtually stationary at great distances for these temperature ranges; thus, the change in strontium void concentration was limited to one unit cell on each side of the boundary. Therefore, a limited point defect approach was used in this paper, only for electrons and strontium vacancies [72]. This, combined with the high density of the defect state interface, explains the huge change in sensor resistivity observed. The authors proposed a point defect model involving the formation of a space charge area near the phase boundary, which has a significant impact on the balance of local defects. The consequence of this is the decisive influence of the overall value on the concentration of each type of defect. The reason for the rapid response of the sensor could then be an increase in the concentration of the cationic void near the interface, induced by the space charge. Cationic voids are the cause of high electron depletion [70]. In the same year, Hu et al. presented a low-temperature nano-structured SrTiO₃ thick-film oxygen sensor obtained by utilization of the high-energy ball milling technique in conjunction with the screen printing technique. At that time, the novelty of the proposed SrTiO₃-based gas sensor was the ability to work at an operating temperature as low as 40 °C, and, in fact, this is a good result nowadays. The sensors were tested under exposure to 2–20% oxygen. Moreover, the experimental results showed that the sensing property of the synthesized SrTiO₃ sensors with an annealing temperature of 400 °C is much better than the commercial SrTiO₃ sensors (both milled and not milled materials) [73]. The effects of the annealing temperature on the sensing properties of nano-sized SrTiO₃ oxygen gas sensors were analyzed by the same group and presented in 2005 [31]. The authors used various methods and techniques to validate the annealing temperature influence, for example, by using differential thermal analysis (DTA)/thermogravimetric analysis (TGA), XRD and transmission electron microscopy (TEM) methods. The films were annealed in the range 400–800 °C with 100 °C steps, and no annealed sample was tested. However, surprisingly, the results showed that samples annealed at 400 °C, as previously shown [73], exhibited the highest sensor response to oxygen at the same operating temperature, 40 °C [31]. The authors concluded that the different annealing temperatures only affect the grain size of the synthesized SrTiO₃-based oxygen sensors, but they do not provide evidence as to why 400 °C showed the highest responses. More interesting is that the results showed that the lower annealing temperature offers higher responses, but the authors did not

test, or at least did not present, the results for samples annealed at 300 °C [31]. The same group in 2004 presented the investigation results on the same gas-sensing material for near-human body temperature oxygen sensing application [74]; however, the presented results did not bring anything new [74]. An effect of oxygen cross-sensitivity was evaluated by Sahner et al. [75]. In 2006, the authors fabricated a hydrocarbons sensor for exhaust gases based on semiconducting doped $SrTiO_3$ for on-board diagnosis [75]. A multilayer resistive sensor based on catalytically activated and non-activated $SrTiO_3$ was tested under various scenarios that may occur during the on-line analysis of exhausting gases in the automotive industry, such as the presence of ethane and propane (as hydrocarbon species) and of hydrogen, carbon dioxide and nitric oxide (interfering gases). The results showed that the non-activated sensor part strongly responds to any reducing gas, whereas the catalytically activated part only detects a slight variation in the oxygen equilibrium concentration. On this occasion, the authors concluded that the actual influence of the platinum content on oxygen sensitivity, temperature dependency and long-term stability needs to be investigated, as well as the cross-sensitivity to different types of hydrocarbons with respect to chain length, presence of unsaturated bindings and aliphatic or aromatic chains [75]. Hydrocarbon sensing results were presented by the same group in 2007 [76] with utilization of a nanoscaled $SrTi_{1-x}Fe_xO_{3-\delta}$-based sensor, where two novel synthesis methods, at that time, were validated, i.e., electrospinning and electrospraying. Moreover, the authors proposed a mechanistic model to explain the impact of the enhanced surface-to-volume ratio of the p-type $SrTi_{1-x}Fe_xO_{3-\delta}$ gas-sensing films. The sensors were investigated in the temperature range from 350 to 450 °C under exposure to propane, propene, hydrogen, NO and CO. The sensors showed a fast, reversible and reproducible response towards propane and propen, with cross-sensitivity towards NO. Neither CO or $H_2$ exposure led to the response changes measured as the resistance ratio [75]. In 2009, Menesklou et al. [77] discussed the effect of impurities on the gas-sensing properties of thin layers of strontium titanate. The admixture of lanthanum (La) in a small concentration significantly influenced the dependence of the electrical conductivity of the material on the oxygen partial pressure. Pure $SrTiO_3$ was characterized by an ambiguous dependence of the electrical conductivity on the aforementioned partial pressure of oxygen, but after adding a small amount of La, this relationship became unambiguous. Samples obtained in this way have response times in the millisecond range, regardless of the thickness of the layers. The reason for this lies in the surface conductivity mechanism. The authors showed that the dependence of $SrTiO_3$'s work can be largely leveled in a small range of oxygen partial pressures by doping an acceptor at a high concentration, in this case, iron. The conductivity of doped strontium titanate with a donor is based on the boundary conductivity of the grains, even at temperatures reaching 1000 °C. The doping concentration affects the much slower kinetics of the bulk equilibrium through long-term drift. The grain size has no effect on the bulk equilibrium kinetics. The authors concluded that the fine-grained strontium titanate polycrystals doped with a small amount of donors should increase the surface-to-weight ratio of the sensor and thus improve its properties. At the same time, there is no acceleration of the drift kinetics, as in the case of donor-doped barium titanate, for example. One of the results of the work was a proposal of a thick-film oxygen sensor (lean combustion) for the control of internal combustion engines, characterized by a fast reaction, low-temperature dependence and sensitivity to oxygen [77].

$SrTiO_3$-based gas sensors have been used not only to detect oxygen but also to detect ozone gas ($O_3$). In 2013, Mastelaro et al. [78] reported the nanocrystalline $SrT_{1-x}Fe_xO_3$ (STF) gas-sensing material for ozone detection. An electron-beam physical vapor deposition (PVD) system was utilized for the deposition with the post-processing annealing process (550 °C). The authors analyzed the effect of iron doping (0.0075, 0.10 and 0.15 mol% of iron) on the gas-sensing characteristics. The sensors exhibited a very high sensor response ($R_0/R$), i.e., 580 under exposure to 0.6 ppm of ozone at 190 °C. The limit of detection was as low as 0.075 ppm [78]. Kajale et al. [79] also showed interesting results of $SrTiO_3$-based gas sensors; the fabricated $SrTiO_3$-based sensors exhibited high sensitivity to CO with no response to

$H_2S$ and vice versa for Cu-doped $SrTiO_3$, which can be explained due to conversion of Cu into CuS. However, this conference paper provides only preliminary results without an extensive elaboration [79]. On the contrary, high responses to $H_2S$ by utilization of pure ST films have been reported by the same group [80]; in this paper, the $SrTiO_3$ nanomaterial was synthesized using a sol–gel hydrothermal method and exhibited an outstanding gas sensor response (S > 500) at a lower temperature, 150 °C, to 80 ppm of $H_2S$ [80]. The possibility to detect another carbon oxide, $CO_2$, was verified by Zaza et al. [81], in 2015. The authors presented the results of a room-operated $SrTiO_3$-based sensor. However, because of the low recovery rate, the regeneration of the sensor has to be made at high temperature for promoting the decomposition of the carbonates formed on the perovskite surface, which significantly reduce the advantage of operation at room temperature [81]. ST-based gas sensors have also been tested under exposure to volatile organic compounds (VOC), such as ethanol. Recently, Trabelsi et al. [82] presented the ethanol-sensing response of a $SrTiO_{3-g}$-based gas sensor fabricated by a solid-state reaction enhanced in a thermally activated process. The electrical properties of the sensors were examined using impedance spectroscopy in a wide range of potential operating temperatures from −33 to 40 °C. The obtained $SrTiO_{3-\delta}$ was characterized by low resistivity and low dielectric losses. The gas-sensing material revealed low-frequency relaxation processes and electrical conductivity resulting from the first ionization of oxygen vacancies. The sensors exhibited a decrease in conductivity after introducing ethanol gas (p-type semiconductor effect). Interesting is the fact that the optimal working temperatures for the tested samples are lower, about 350–370 °C, which suggests the dominant role of oxygen vacancies in detecting the presence of ethanol [82].

Apart from the conventional resistive-type $SrTiO_3$-based gas sensors, ST was used as a gas-sensitive layer for fiber optic evanescent-wave hydrogen sensors, where $SrTiO_3$ films were doped with lanthanum (La) [83]. These sensors show a rapid, reproducible sensing response to hydrogen fuel gas streams at elevated temperatures (600–800 °C). The presence of hydrogen results in a reversible and reproducible decrease in near-infrared transmission through the sensor. Sensors were also tested directly in the anode assembly of an operating solid oxide fuel cell (SOFC) with the sensor response correlating with both $H_2$ concentration and SOFC cell potential. Sensors based on optical fiber platforms are well suited for harsh environment sensing applications and are safer than traditional resistive sensing technologies for combustible environments, showing the absence of an electrical current and therefore removing the risk of electrical interference to or from the operating fuel cell [83]. ST has been used as a gas-sensing material for microwave-based gas sensors since 2007, when Jouhannaud et al. [84] presented a coaxial structure operated in the 0.3 MHz–3 GHz frequency range under exposure to various species such as ethanol, the saturated vapor of water and toluene. The response of the sensor is quantitative and typical for each gas. This method of measurement leads to the development of an alternative to the current gas sensor. It has to be underlined that these results opened new research in gas-sensing measurements, i.e., operating at the microwave frequency range [85–88]. The examples of the $SrTiO_3$-based gas sensors are presented in Figure 4.

Table A1 in Appendix A summarizes the information on the $SrTiO_3$ sensor properties tests performed so far. The first column contains the material (chemical formula), the second one contains information about the operating temperature, the next one with the maximum response and in the following target gas, the sample thickness (if the authors provided such information), the deposition method and the literature reference in the last column. Table A1 summarizes the information contained in this section.

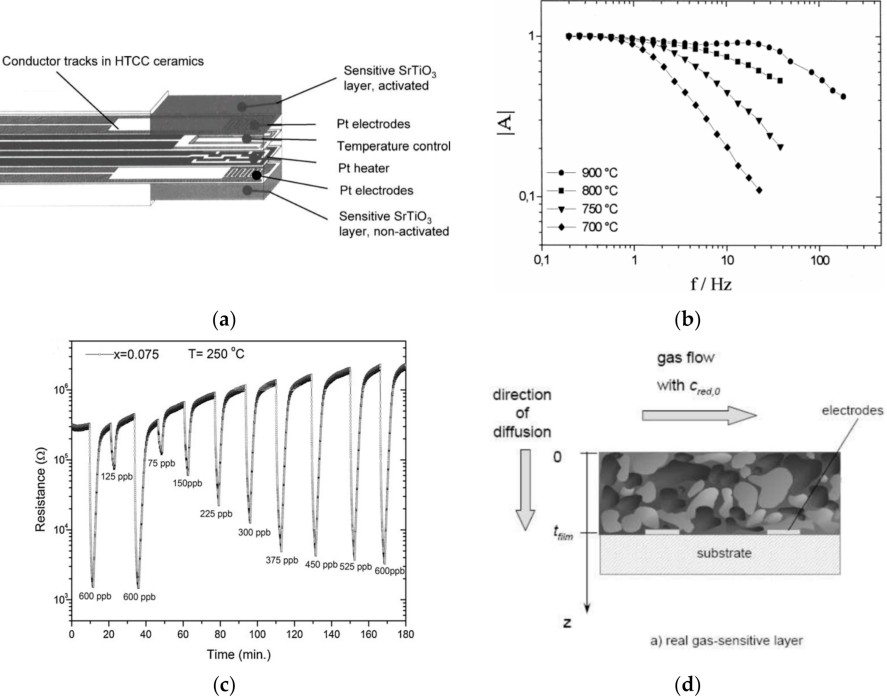

**Figure 4.** Examples of SrTiO$_3$-based gas sensors: (**a**) Sketch of the sensor structure fabricated in the HTCC (high-temperature cofired ceramics) technology. Alumina tapes were used as the substrate with SrTiO$_3$ doped with 1 at.% Ta as the gas-sensitive layer. Reprinted with permission from [89] Copyright 2021 Elsevier. (**b**) Magnitude |A| of a SrTi$_{0.65}$Fe$_{0.35}$O$_3$ thick-film sensor with dependence on the frequency of the oxygen partial pressure. This experiment was carried out to investigate the kinetic behavior of SrTi$_{0.65}$Fe$_{0.35}$O$_3$. At 900 °C, response times of 10 ms can be achieved. Reprinted with permission from [77] Copyright 2021 Elsevier. (**c**) shows the graph of SrTi$_{0.925}$Fe$_{0.075}$O$_3$ as a function of ozone concentration and operating temperature (250 °C). The measurement cycles consisted of the ozone exposure time—2 min. The sample looks like a p-type semiconductor (the resistance of the sample decreases as oxidizing gases are absorbed). The diagram shows that the sample responds well to O$_3$, from a concentration of 75 ppb O$_3$. On the other hand, it is saturated at a concentration higher than 525 ppb of ozone. The saturation effect is due to the limited number of absorption sites. Reprinted with permission from [78] Copyright 2021 Elsevier. (**d**) A simplified one-dimensional approach to modeling screen-printed thick-film sensors. The sensor is modeled in two steps, macroscopic and microscopic. The macroscopic part describes the gas transport through a gas-sensitive porous film. In their work, the authors calculated the appropriate profile of bracing for the one-dimensional geometry shown in the figure.

### 2.2. BaTiO$_3$ for Gas-Sensing Applications

In 2004, K. Park and D.J. Seo [90] published a paper on the CO detection characteristics of BaTiO$_3$-based gas sensors with graphite doping. It was found that the difference in resistivities measured at high temperatures (400 °C) in the air and CO gas atmosphere (5–100%) change with increasing graphite content. The porosity of the material also increases, which leads to an improvement in CO gas detection sensitivity. Greater porosity provides more sites on the sample surface for oxygen adsorption and the reaction between CO and O$^-$ gas, and between CO gas and O. The gas-sensing behavior for CO$_2$ with the utilization of BST-based gas sensors has been studied as well, for example, by the utilization of BST and CuO [91], PbO [92] and LaCl$_3$ [93]. In 2006, Mandayo, Gemma García et al. [94] developed a thin sensor film to detect CO$_2$. The authors focused on response rates for various temperatures, target gases and relative humidity concentrations for indoor air quality (IAQ) monitoring.

Various attempts have been made to improve the selectivity and sensitivity of BaTiO$_3$-based sensors using dopants and additives. For example, La has been proposed [95]

to increase the response for $NO_2$ and $NH_3$ at room temperature. G.N. Chaudhary et al. [96] proposed thin-film $BaTiO_3$ doped with CuO and CdO for liquefied petroleum gas (LPG) detection. The study showed that doping with various concentrations of CuO/CdO influenced the sensitivity. In addition, incorporation of a 0.3 wt.% Pd-doped $CuO:CdO:BaTiO_3$ element showed high sensitivity with selectivity to the other gases including CO, $H_2$ and $H_2S$. In [62], the authors investigated the influence of the band gap, size and shape of a BT thin-film on the gas-sensing mechanism of $H_2S$. The obtained sensors exhibited the highest response at 300 °C to 200 ppm of the target gas. Gas-sensing characteristics under exposure to $H_2S$ were measured by Huang, He-Ming, et al. [97]. In 2007, the authors reported that $Ba_{0.99}Ce_{0.01}TiO_3$ sensors doped with $Fe_2O_3$ achieved response of 2.91 towards 0.4 ppm $H_2S$ at 150 °C (the sensor response is defined as $R_{N2}/R_{H2S}$, where $R_{N2}$ and $R_{H2S}$ are the electrical resistances of the sensor in $N_2$ and $H_2S$, respectively). Additionally, fast response–recovery dynamics were obtained with 40 and 55 s for response and recovery times, respectively. The advantages of the above-mentioned BT composition can be summarized as a high response, lower operating temperature and fast response–recovery times, which make this material very attractive for industrial application. However, a long-term stability test has not been conducted to confirm these assumptions. $BaTiO_3$-based gas sensors have also been applied as humidity detectors. In [98], the authors presented results on $BaTiO_3$ nanoframes and $TiO_2$-$BaTiO_3$ nanotubes tested under various humidity concentrations (the samples were tested at RH from 15% to 95% at room temperature). The obtained results showed a high and reversible response towards different water concentrations. As a result, these sensors possess high stability, fast response times and reproducibility and could be used as humidity sensors.

The examples of $BaTiO_3$-based gas sensors are presented in Figure 5. Table A2 in Appendix A is a summary of the information contained in the chapter on $BaTiO_3$ sensor properties. It contains information on the test substance, operating temperature, maximum response, target gas, sample thickness, deposition method and references to papers by other authors.

### 2.3. BaSrTiO₃ for Gas-Sensing Applications

In 2000, X.F. Chen et al. [99] studied the hydrogen gas sensitivity of sputtered amorphous $Ba_{0.67}Sr_{0.33}TiO_3$ thin films. The BST material was prepared by using the RF magnetron co-sputtering process (200 nm, 300 °C, 1.8 Pa, 50% $O_2$). The testing was carried out at one atmosphere in a measurement chamber with 500 sccm of target gas flow (1042 ppm of $H_2$) in the operating temperature range from 80 to 250 °C. The authors concluded that higher sensitivity (defined as the ratio of the dc current, $I_{gas}/I_{air} \approx 7.3$) to $H_2$ can be obtained in the temperature range of 170–190 °C. In another work [100], the authors proposed thin films of Cu-doped $Ba_{0.75}Sr_{0.25}TiO_3$ for $H_2S$ detection. The gas-sensing layers were synthesized under hydrothermal conditions and further deposited via the RF sputtering technique onto $Al_2O_3$ substrates provided with Au electrodes and a Pt heater. The relative humidity influence (5%, 30%, 50%, 70%) on changes in the electrical resistance of BST with 5% Cu exposed to different concentrations of $H_2S$ (5, 10, 30, 50, 70, 90 ppm) at operating temperatures in the range from 100 to 400 °C was investigated. The authors reported that at 250 °C, 50% RH and 10 ppm of $H_2S$ concentration, Cu 5% BST showed good stability in electrical resistance: $R_{air} = 4.25 \cdot 10^{10} \pm 20\%$ Ω, $R_{H2S} = 5.61 \cdot 10^8 \pm 13.7\%$ Ω (response $S = R_{air}/R_{H2S}$, S = 75.76). Recently, results on the gas-sensing properties of BST: $Ba_{0.5}Sr_{0.5}TiO_3$ (BST0.5) and $Ba_{0.7}Sr_{0.3}TiO_3$ (BST0.7) thin films under $H_2S$ exposure (800 ppm) in the temperature range of 50–380 °C were presented in [70]. The gas sensors based on both BSTs, i.e., 0.5/0.5 and 0.7/0.3, reached the maximum response (57.57% and 41.61% for $BST_{0.5}$ and $BST_{0.7}$, respectively) at 330 °C. BST-based nanomaterials have also found application in the detection of $NH_3$ gas [101–103]. In 2005, Somnath C. Roy et al. [101] proposed novel ammonia-sensing phenomena of $Ba_{0.5}Sr_{0.5}TiO_3$ thin films obtained by the sol–gel spin coating technique. The deposited sensors were tested under exposure to different ammonia concentrations (160, 320, 640 and 1280 ppm). The sensitivity

(S) was defined as $S = ((R_g - R_{air})/R_{air})$, where $R_{air}$ and $R_g$ are the electrical resistances in air and the target gas, respectively. The obtained results showed that sensitivity $S$ was found to be around 20% when thin films were exposed to $NH_3$ at a concentration of 160 ppm and increased to 60% when the concentration was raised to 1280 ppm. The researchers tested BST thin films for cross-sensitivity to other gases such as CO, $NO_2$ and ethanol. The obtained results showed no detectable resistance change when operated under conditions similar to those of ammonia sensing. $NH_3$-sensing characteristics were measured at room temperature as well [69]. $Ba_{0.75}Sr_{0.25}TiO_3$-based gas sensors were tested in the range of 30–110 ppm of ammonia with various relative humidity concentrations (10%–70%) at room temperature (23 °C). The authors defined the sensor signal S, calculated as the ratio of $S = (R_{air}/R_{gas})$, where $R_{air}$ is the resistance in the humid background and $R_{gas}$ is the resistance in the presence of $NH_3$ dosed in the humid background. During these measurements, the humid background was set as 10%, 30%, 50% and 70%. The highest sensor response ($S \approx 2.5$) was obtained for the 110 ppm concentration of $NH_3$ and 50% RH. Therefore, the effect of the humidity influence opened the possibility to use BST gas sensors as humidity sensors [104–106]. For example, in [106], the temperature dependence of conductance and susceptance versus relative humidity (20–80%) for MgO-doped BST in the temperature range 10–60 °C was measured. The authors proposed the reaction mechanism model based on the impedance measurements, where a parallel combination of a resistor and a capacitor was used. Further work [104] focused on the influence of MgO doping (3% mol) on the structural and perceptible properties of BST0.5 in the humidity range of 20–95% in comparison to non-doped BST materials. The authors reported that the sensor including BST0.5 added with 3 mol% MgO exhibited improved humidity sensitivity and showed a faster response than pure BST0.5.

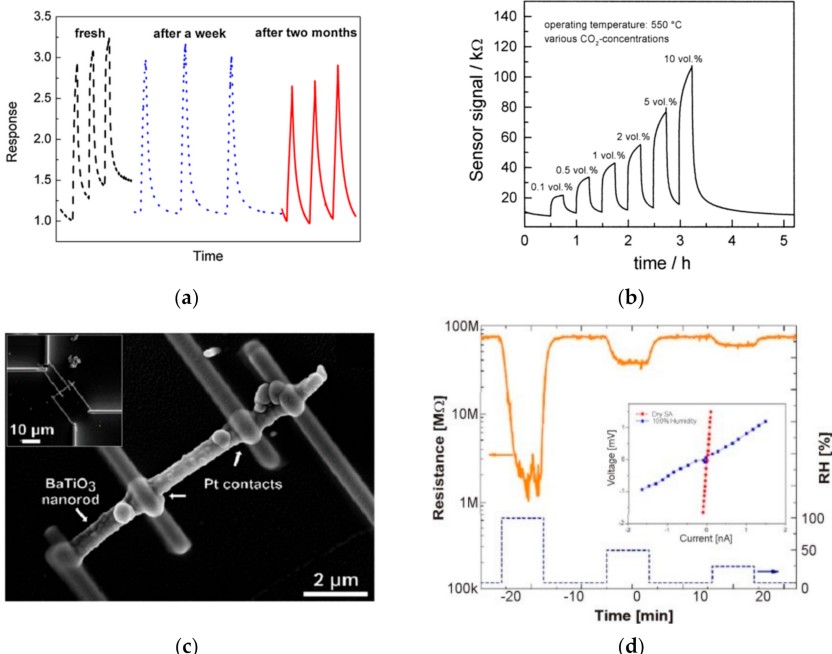

**Figure 5.** Examples of $BaTiO_3$ gas sensors: (**a**) Stability and repeatability of the $Fe_2O_3$-$Ba_{0.99}Ce_{0.01}TiO_3$ sensor up to 400 ppb $H_2S$ at 150 °C. To check the repeatability and stability of the sensor, tests were carried out on a fresh sensor, after one week and two months. The sensor was tested by alternating cycles of 400 ppb $H_2S$ and $N_2$ at 150 °C. The tested sensor showed quite good stability and repeatability. The longer operation of the sensor at the temperature of 150 °C resulted in the disappearance of the upward drift of the base resistance, which contributed to the improvement of the sensor's performance. Reprinted with permission from [97] Copyright 2021 Elsevier. (**b**) Sensor response of the gas-sensing material based on $BaTiO_3$ and $CuO$ (1:1 molar ratio) with 10 wt. $LaCl_3$ to various $CO_2$ concentrations in the range 0.1–10 vol.%. at the operating temperature of 550 °C. Reprinted with permission from [93] Copyright 2021 Elsevier. (**c**) Detail of the $BaTiO_3$ nanoprobe in contact with FIB nanolithography in a four-probe configuration. The insert in the upper left corner shows a low-magnification image of the same device. The platinum strips deposited with focused ion beam lithography (FIB) are clearly shown. The figure shows a prototype device created by integrating $BaTiO_3$ nanorods using FIB nanolithography. The four-probe electrical measurements on the individual $BaTiO_3$ nanorods on this sample showed resistivity values from 10 to 100 $\Omega$cm, which is the typical value range for oxygen-deficient $BaTiO_3$. (**d**) Scalable and reproducible response of a nanosensor sample made of $BaTiO_3$ nanorods at room temperature to changes in air humidity (RH), with different responses to different levels of air humidity (100, 50 and 25%) in synthetic air. The inset shows the I–V curves obtained in dry and humid (100% RH) air.

The examples of $BaSrTiO_3$-based gas sensors are presented in Figure 6. A collective summary of the information contained in this chapter, on the sensor properties of $BaSrTiO_3$, can be found in Table A3 in Appendix A, it contains basic information, such as: chemical formula of the tested compound, operating temperature, maximum response, target gas, information on sample thickness, deposition method and references to literature sources.

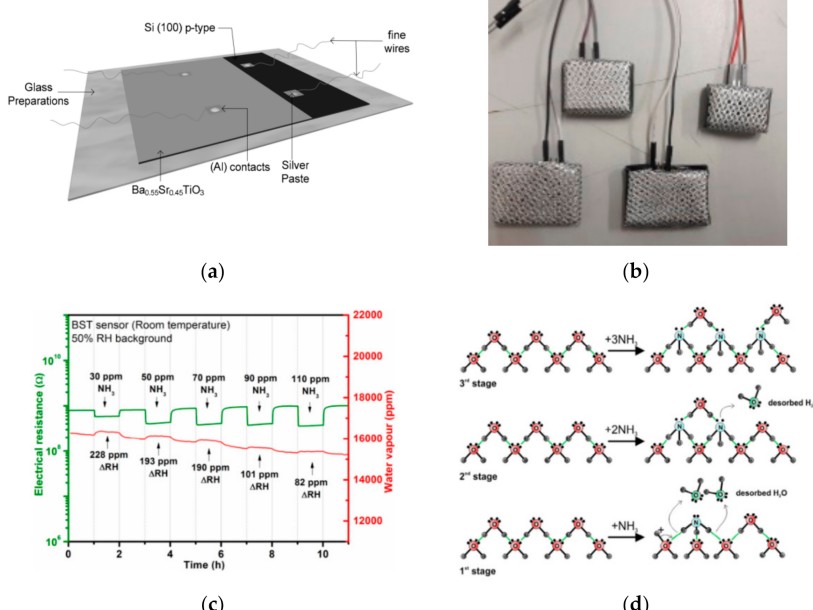

(a)  (b)

(c)  (d)

**Figure 6.** Examples of BaSrTiO$_3$ gas sensors: (**a**) The Ba$_{0.5}$Sr$_{0.5}$TiO$_3$ film doped with RuO$_2$ model (the result of the copper wire installation) of a sensor for bad breath gas analysis. (**b**) The physical appearance of a Ba$_{0.5}$Sr$_{0.5}$TiO$_3$ film doped with a RuO$_2$ probe of a sensor for bad breath gas analysis. (**c**) The gas-sensing response of the BaSrTiO$_3$ gas sensor under exposure to NH$_3$ (30–110 ppm) at various RH concentrations. Reprinted with permission from [69] Copyright 2021 Elsevier. (**d**) Sketch of the gas reaction model of BST-based gas sensors to various NH$_3$ concentrations. This model showed the impact of RH capillary condensation towards NH$_3$ detection. Reprinted with permission from [69] Copyright 2021 Elsevier.

*2.4. BaSrTiO$_3$ for NO$_2$ Detection*

The utilization of BST-based gas sensors for nitrogen dioxide detection was previously studied by [107], as mentioned in Section 2.3. For example, in [100], the authors showed that the electrical resistance of BaSrTiO$_3$ doped with 5% Cu increases during the exposition to NO$_2$ at 3 ppm. Therefore, it could be suggested that BST Cu 5% behaves as an n-type semiconductor. However, in Figure 6, we present the results obtained for BST doped with Cu, where the electrical resistances decrease when NO$_2$ was introduced to the measurement chamber. The BST-Cu gas-sensitive layer was deposited by the utilization of magnetron sputtering deposition technology. Firstly, the BST as a base material was deposited from the Ba$_{0.6}$Sr$_{0.4}$TiO$_3$ homemade target, recently presented in [108], and then copper dopants were deposited. The deposition system for co-sputtering was previously presented in [109]. The pure BST-based gas sensor had base resistance around 5–6 GOhm, while after the Cu modification, the electrical resistance reduced to 8–9 MOhm at RT and the gas-sensing effect was achieved. Figure 7a shows the sensor response defined as $R_0/R_g$, where $R_0$ and $R_g$ are electrical resistances measured in the synthetic air and target gas, respectively. The gas-sensing measurement setup was previously presented in [110]. As can be noticed, the highest responses were obtained around 250 °C, and therefore this temperature was set as the operating temperature. The NO$_2$-sensing effect was measured in the 0.4–20 ppm range at a constant operating temperature (Figure 7b) and various RH concentrations (Figure 7c). As can be observed, the sensor reached the maximum response at 10 ppm, and further increasing the NO$_2$ concentration did not change the response. However, due to the measurement setup limitations, the lowest NO$_2$ concentration was 0.4 ppm, but it seems that the obtained sensors could detect lower concentrations, and the theoretical limit of detection was around 0.01 ppm. Finally, a multitest was conducted, where constant NO$_2$ (20 ppm) and RH concentrations (30%, 50%, 70%) were kept to verify the stability. As is shown in Figure 7d, slight changes can be observed (~5%); however, what is interesting is

that the electrical resistances decrease, suggesting p-type behavior, contrary to the results presented by Stanoiu et al. [100]. Therefore, additional measurements are needed.

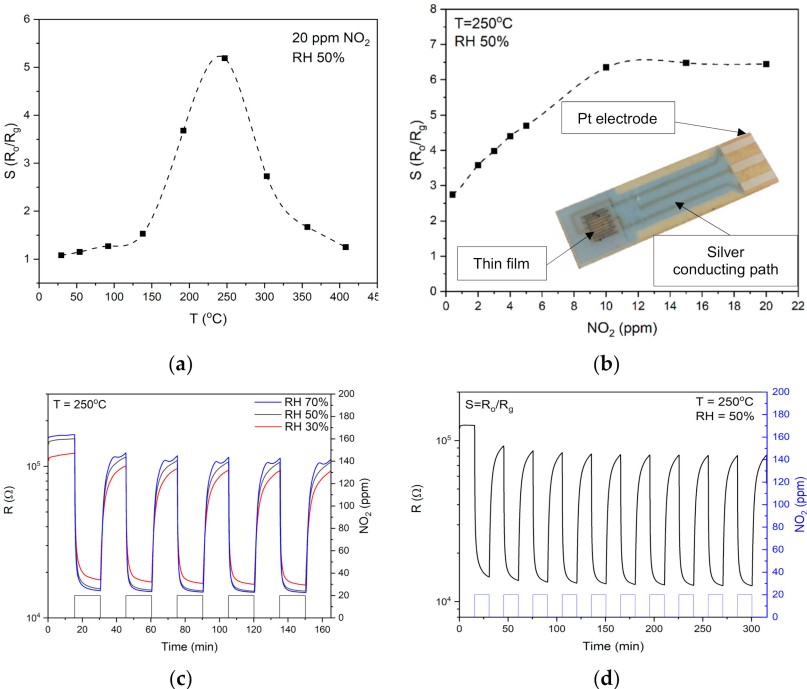

**Figure 7.** The gas-sensing characteristics obtained for a BST-Cu-based gas sensor: (**a**) the sensor response in the wide temperature range under exposure to 20 ppm of $NO_2$ and 50% of RH; (**b**) the sensor response in the 0.4–20 ppm range at constant operating temperature and 50% of RH (inset: a photo of the fabricated gas sensor); (**c**) the resistance changes of the developed gas sensors at various RH concentrations and 20 ppm of $NO_2$; and (**d**) the multitest of the gas sensor response measured at 250 °C, 50% RH and 20 ppm of $NO_2$.

## 3. Conclusions

The gas sensor industry is constantly expecting novel materials that allow researchers to develop gas sensors with higher 3S parameters, known as sensitivity, selectivity and stability, while stability plays a crucial role in most industrial applications. Therefore, the materials that are characterized by longer stability are chosen as a first choice for gas-sensing layers, such as strontium titanate, barium titanate and barium strontium titanate.

In this paper, the recently published results on the above-mentioned materials for gas-sensing applications are presented and discussed, including the recently obtained results by the authors of $NO_2$ detection in the automotive application. The BST-Cu sensors exhibited the highest responses around 250 °C and worked well in the 0.4–20 ppm range; however, above 10 ppm, the sensor response achieved a constant value. The gas sensor response increased when relative humidity increased. At the same time, the sensor exhibited good repeatability, proven by the multitest, which confirms the possibility to utilize BST-Cu gas sensors in industrial applications.

**Funding:** The research leading to these results received funding from the Norway Grants 2014–2021 via the National Centre for Research and Development.

**Institutional Review Board Statement:** Not applicable.

**Informed Consent Statement:** Not applicable.

**Acknowledgments:** The work was carried out in the Biomarkers Analysis Laboratory AGH at the Institute of Electronics AGH.

**Conflicts of Interest:** The authors declare no conflict of interest.

# Appendix A

**Table A1.** The summaries of $SrTiO_3$-based gas sensors.

| Material | Operating Temp. | Max. Response | Target Gas | Thickness | Deposition Method | Reference |
|---|---|---|---|---|---|---|
| $SrTiO_3$ | 250–350 °C | 0–34 [1] | $C_2H_5OH$ | 400 nm | sol–gel | [111] |
| $SrTiO_3$ | 40 °C | 6 [2] | $O_2$ | 45 μm | high-energy ball milling technique and thick-film screen printing technique | [31,73] |
| $SrTi_{0.925}Fe_{0.075}O_3$ | 250 °C | 170–580 [2] | $O_3$ | 70 nm | polymeric precursor method | [78] |
| $SrTi_{0.9}Fe_{0.1}O_3$ | 250 °C | 10 [2] | $O_3$ | 70 nm | polymeric precursor method | [78] |
| $SrTi_{0.85}Fe_{0.1}5O_3$ | 250 °C | 53 [2] | $O_3$ | 70 nm | polymeric precursor method | [78] |
| $SrTi_{0.85}Fe_{0.15}O_3$ | 220 °C | 267 [2] | $O_3$ | 70 nm | polymeric precursor method | [78] |
| $SrTi_{0.85}Fe_{0.15}O_3$ | 190 °C | 580 [2] | $O_3$ | 70 nm | polymeric precursor method | [78] |
| $SrTiO_3$ | 350 °C | 27 [2] | CO | thick film | screen-printed on glass substrate in the desired pattern | [79] |
| $SrTiO_3$ | 350 °C | 3 [2] | $H_2S$ | thick film | screen-printed on glass substrate in the desired pattern | [79] |
| $SrTiO_3$ | 350 °C | 18 [2] | $NH_3$ | thick film | screen-printed on glass substrate in the desired pattern | [79] |
| $SrTiO_3$ with modified surface | 300 °C | 7 [2] | $NH_3$ | thick film | screen-printed on glass substrate in the desired pattern | [79] |
| $SrTiO_3$ with modified surface | 300 °C | 22 [2] | $H_2S$ | thick film | screen-printed on glass substrate in the desired pattern | [79] |
| $SrTiO_3$ | RT | 2% [3] | $CO_2$ | thick film | citrate–nitrate combustion synthesis method | [81] |
| $LaAlO_3/SrTiO_3$ | RT, 80 °C | 650, 4000 [4] | $H_2$ | thin film | laser molecular beam epitaxy | [112] |
| $Sr_{0.995}La_{0.005}TiO_3$ | 850 °C | 4–8 [5] | $O_2$ | 780 μm | screen printing technique | [77] |
| $SrTi_{0.65}Fe_{0.35}O_3$ | 750–850 °C | 2 [5] | $O_2$ | 11 μm | screen printing technique | [77] |
| Ta-doped $SrTiO_3$ | 700 °C | 0.6 [6] | Hydrocarbon ethan | 20 μm | high-temperature co-fired ceramic (HTCC) technology | [89] |
| $SrTiO_3$ | 40 °C | 6.35 [2] | $O_2$ | thick film | physical high-energy ball milling technique | [74] |
| $SrTiO_3$ | 150 °C | 550 [7] | $H_2S$ | 65–75 μm | sol–gel hydrothermal method | [80] |
| $SrTiO_3$ | 350 °C | 100 [7] | $C_2H_5OH$ | 65–75 μm | sol–gel hydrothermal method | [80] |
| $SrTiO_{3-\delta}$ ($\delta$ = 0.075 and 0.125) | 350 °C | 16 [1] | $C_2H_5OH$ | 2 mm | the conventional solid-state reaction method | [82] |
| $SrTiO_3$ | RT | 41 [6] | $O_2$ | 50 nm | pulsed laser deposition (PLD) | [28] |
| 0.14 at.% Nb:$SrTiO_3$ | RT | 1.5 [6] | $O_2$ | 50 nm | pulsed laser deposition (PLD) | [28] |
| $SrTiO_3$ | RT | 1000 [6] | $O_2$ | 20 nm | atomic layer deposition (ALD) | [113] |
| $SrTiO_3$ | RT | 500 [6] | $O_2$ | 20 nm | superposition and patterning | [114] |
| $SrTi_{0.8}Fe_{0.2}O_{3-\delta}$ | 400 °C | 12 [6] | $C_3H_8$ | 10–20 μm | screen printing technique | [75] |
| $SrTi_{0.8}Fe_{0.2}O_{3-\delta}$ | 400 °C | 4.2 [6] | $NH_3$ | 10–20 μm | screen printing technique | [75] |
| $SrTi_{0.8}Fe_{0.2}O_{3-\delta}$ | 400 °C | 2.4 [6] | NO | 10–20 μm | screen printing technique | [75] |
| $SrTi_{0.8}Fe_{0.2}O_{3-\delta}$ | 400 °C | 2.4 [6] | CO | 10–20 μm | screen printing technique | [75] |
| $SrTi_{0.8}Fe_{0.2}O_{3-\delta}$ | 400 °C | 1.6 [6] | $H_2$ | 10–20 μm | screen printing technique | [75] |

[1] $(R_g - R_0)/R_0$—in the paper, the authors used sensitivity calculated from the change in resistance, not a response, unit: a.u. [2] $R_0/R_g$—Max. Response—the ratio of the sensor resistance without gas and in the presence of gas, unit: a.u. [3] $(C_0 - C)/C_0 \cdot 100\ \%$—in the paper, the authors used sensitivity calculated from the change in capacity, unit: %. [4] $((I_0 - I)/I_0) \cdot 100\ \%$—in the paper, the authors used sensitivity calculated from the change in current, unit: %. [5] $G_0/G_g$—Max. Response—the ratio of the sensor conductivity without gas and in the presence of gas, unit: a.u. [6] $R_g/R_0$—Max. Response—the ratio of the sensor resistance in the presence of gas and without gas, unit: a.u. [7] $(G_g - G_0)/G_0$—in the paper, the authors used sensitivity calculated from the change in conductance, not a response, unit: a.u.

**Table A2.** The summaries of BaTiO$_3$-based gas sensors.

| Material | Operating Temp. | Max. Response | Target Gas | Thickness | Deposition Method | Reference |
|---|---|---|---|---|---|---|
| BaTiO$_3$ | 25 °C | 50 [1] | humidity | 750 nm | sol–gel processing | [115] |
| BaTiO$_3$ | 300 °C | 300 [2] | H$_2$S | thick film | low-temperature hydrothermal route | [62] |
| BaTiO$_3$ | RT | 116 [3] | H$_2$O$_2$ | 1 µm | electroless deposition method | [116] |
| PVDF-BaTiO$_3$ composite | RT | 0.2416 [4] | humidity | 150 nm | mixing nanoparticles | [117] |
| Modified BaTiO$_3$ | 350 °C | 1119 [2] | H$_2$S | 65–70 µm | screen printing technique | [118] |
| Modified BaTiO$_3$ | 350 °C | 31 [2] | NH$_3$ | 65–70 µm | screen printing technique | [118] |
| BaTiO$_3$ | 350 °C | 303 [2] | LPG | thin film | spray pyrolysis techniques | [119] |
| Ce-doped BaTiO$_3$ | 200 °C | 2.99 [5] | H$_2$S | 500 nm | coprecipitation method | [97] |
| Fe$_2$O$_3$-Ba$_{0.99}$Ce$_{0.01}$TiO$_3$ | 150 °C | 4.36 [5] | H$_2$S | 500 nm | coprecipitation method | [97] |
| Fe$_2$O$_3$-Ba$_{0.99}$Ce$_{0.01}$TiO$_3$ | 150 °C | 2.91 [5] | H$_2$S | 500 nm | coprecipitation method | [97] |
| BT-NH$_2$ NPs | RT | 5 [6] | cysteine | 50 µm | solid-state reaction method | [120] |
| BaTiO$_3$ | 280 °C | 2.9 [7] | CO$_2$ | thick film | screen printing technology | [121] |
| BaTiO$_3$ | 200 °C | 10 [8] | CO$_2$ | 55 nm | RF sputtering | [122] |
| BaTiO$_3$ | RT | 17.6 [9] | NO$_x$ | 350 nm | sol–gel spin coating method | [123] |
| TiO$_2$-BaTiO$_3$ nanotubes | RT | 22.8 [10] | humidity | nanorods | sol–gel electrophoretic deposition technique, electrochemical anodization technique | [98] |
| BaTiO$_3$-CuO | 300 °C | 1.8 [7] | CO$_2$ | 1 µm | RF sputtering technique | [94] |
| BaTiO$_3$-CuO | 300 °C | 0.1 [11] | CO$_2$ | 1 µm | RF sputtering technique | [94] |
| BaTiO$_3$-CuO | 400 °C | 1.12 [7] | CO$_2$ | 1 µm | RF sputtering technique | [94] |
| BaTiO$_3$-CuO | 400 °C | 0.7 [11] | CO$_2$ | 1 µm | RF sputtering technique | [94] |
| CuO-BaTiO$_3$ Ag doped | 430 °C | 1.59 [7] | CO$_2$ | thick film | conventional sintering method | [124] |
| BaTiO$_3$ La doped | 950 °C | 0.75 [7] | O$_2$ | 1 µm | magnetron sputtering | [125] |
| BaTiO$_3$ | RT | 22.8 [10] | humidity | nanorods | sol–gel electrophoretic deposition | [126] |
| BaTiO$_3$ | 25 °C | 70 [12] | humidity | nanofiber | electrospinning | [127] |
| CaO-BaTiO$_3$ | 160 °C | 0.68 [13] | CO$_2$ | thick film | typical synthesis procedure | [128] |
| BaTiO$_3$-CuO-La$_2$O$_3$ | 400 °C | 1.05 [7] | CO$_2$ | thick film | screen printing technique and firing | [93] |
| BaTiO$_3$-CuO-LaCl$_3$ | 400 °C | 1.25 [7] | CO$_2$ | thick film | screen printing technique and firing | [93] |
| BaTiO$_3$-LaCl$_3$ | 400 °C | 1.55 [7] | CO$_2$ | thick film | screen printing and firing | [93] |
| Ba$_{0.999}$Sb$_{0.001}$TiO$_3$ | 400 °C | 15 [7] | CO | 1 mm | adding graphite powders | [90] |
| BaTiO$_3$ | 25 °C | 3.5 [7] | LPG | thin film | sol–gel method | [129] |

[1] $Z_{90\%}/Z_{40\%}$—Max. Response—sensor impedance ratio at 90% and 40% air humidity, unit: a.u. [2] $(G_g-G_0)/G_0$—in the paper, the authors used sensitivity calculated from the change in conductance, not a response, unit: a.u. [3] the response current density unit: µA·cm$^{-2}$. [4] unit: pF/%RH. [5] $R_0/R_g$—Max. Response—the ratio of the sensor resistance without gas and in the presence of gas, unit: a.u. [6] $I/I_0$—Max. Response—the ratio of the sensor current without cysteine and in the presence of cysteine, unit: a.u. [7] $R_g/R_0$—Max. Response—the ratio of the sensor resistance in the presence of gas and without gas, unit: a.u. [8] $((I_0-I)/I_0)\cdot100$ %—in the paper, the authors used sensitivity calculated from the change in current, unit: %. [9] $(R_0-R_g)/R_0$—in the paper, the authors used sensitivity calculated from the change in resistance, not a response, unit: a.u. [10] $R_{0\%}/R_{100\%}$—Max. Response—sensor resistance ratio at 0% and 100% air humidity, unit: a.u. [11] $C_g/C_0$—Max. Response—the ratio of the sensor capacitance without gas and in the presence of gas, unit: a.u. [12] $Z_{11\%}/Z_{95\%}$—Max. Response—sensor impedance ratio at 11% and 95% air humidity, unit: a.u. [13] $(R_g-R_0)/R_0$—in the paper, the authors used sensitivity calculated from the change in resistance, not a response, unit: a.u.

**Table A3.** The summaries of $BaSrTiO_3$-based gas sensors.

| Material | Operating Temp. | Max. Response | Target Gas | Thickness | Deposition Method | Reference |
|---|---|---|---|---|---|---|
| $BaTiO_3$/ $SrTiO_3$ | 250–350 °C | 10–100 [1] | $C_2H_5OH$ | 400 nm | sol–gel method | [111] |
| Cu mol. 5%-doped perovskite material $Ba_{0.75}Sr_{0.25}TiO_3$ | 250 °C | 400 [2] | $H_2S$ | 386.4 nm | RF sputtering technique | [100] |
| Co-doped $BaSrTiO_3$ | RT | 35 [3] | $CO_2$ | thin film | RF sputtering technique | [130] |
| $Ba_{0.5}Sr_{0.5}TiO_3$ | 270 °C | 60 [1] 40 [1] 30 [1] | $NH_3$ | 150 nm 320 nm 480 nm | sol–gel method | [103] |
| Cu mol. 5%-doped $BaSrTiO_3$ | 200 °C | 15 [2] | $H_2S$ | thick film | hydrothermal method | [68] |
| Cu mol. 0.1%-doped $BaSrTiO_3$ | 150 °C | 5 [2] | $NH_3$ | thick film | hydrothermal method | [68] |
| Sr 0.2-doped $BaTiO_3$ | RT | 100 [4] | $NO_2$ | thick film | low-temperature hydrothermal route | [107] |
| Sr 0.2-doped $BaTiO_3$ | RT | 55 [4] | $NH_3$ | thick film | low-temperature hydrothermal route | [107] |
| Sr 0.2-doped $BaTiO_3$ | 400 °C | 28 [4] | $H_2S$ | thick film | low-temperature hydrothermal route | [107] |
| Sr 0.4-doped $BaTiO_3$ | 400 °C | 2 [4] | LPG | thick film | low-temperature hydrothermal route | [107] |
| $Ba_{0.67}Sr_{0.33}TiO_3$ | 205 °C | 34 [5] | $H_2$ | 200 nm | RF magnetron co-sputtering | [99] |
| $(Ba_{0.87}Sr_{0.13})TiO_3$ | 300 °C | 29 [2] 85 [2] 62 [2] 56 [2] | $NH_3$ | 17 μm 33 μm 48 μm 63 μm | screen printing technique | [101] |
| $(Ba_{0.87}Sr_{0.13})TiO_3$ | 300 °C | 26 [2] | $H_2S$ | 33 μm | screen printing technique | [101] |
| $Cr_2O_3$-modified $(Ba_{0.87}Sr_{0.13})TiO_3$ | 350 °C | 16.6 [2] | $NH_3$ | 33 μm | screen printing technique | [131] |
| $Cr_2O_3$-modified $(Ba_{0.87}Sr_{0.13})TiO_3$ | 350 °C | 73 [2] | $H_2S$ | 33 μm | screen printing technique | [131] |
| $Ba_{0.67}Sr_{0.33}TiO_3$ | 350 °C | 10 [4] | $H_2S$ | 65–70 μm | screen printing technique | [132] |
| Cu-doped $Ba_{0.67}Sr_{0.33}TiO_3$ | 350 °C | 26.5 [4] | $H_2S$ | 65–70 μm | screen printing technique | [132] |
| Cr-doped $Ba_{0.67}Sr_{0.33}TiO_3$ | 350 °C | 46.4 [4] | $H_2S$ | 65–70 μm | screen printing technique | [132] |
| $Ba_{0.998}Sr_{0.002}TiO_3$ | 400 °C | 25 [4] | $H_2S$ | thick film | hydrothermal method | [133] |
| $Ba_{0.998}Sr_{0.002}TiO_3$ in 2% Sn dipping for 20min | 200 °C | 2274 [4] | $H_2S$ | thick film | hydrothermal method | [133] |
| $Ba_{0.5}Sr_{0.5}TiO_3$ | 330 °C | 57.57 [5] | $H_2S$ | thin film | pulse laser deposition (PLD) technique | [70] |
| $Ba_{0.7}Sr_{0.3}TiO_3$ | 330 °C | 41.61 [5] | $H_2S$ | thin film | pulse laser deposition (PLD) technique | [70] |
| $Ba_{0.75}Sr_{0.25}TiO_3$ | RT | 2.5 [2] | $NH_3$ | 50 μm | hydrothermal method | [69] |

[1] $(R_g-R_0)/R_0$—where $R_g$ and $R_0$ are the electrical resistances measured in the presence of gas and synthetic air, respectively. [2] $R_0/R_g$.
[3] $R_g/R_0$. [4] $(G_g-G_0)/G_0$—where $G_g$ and $G_0$ are the electrical conductances measured in the presence of gas and synthetic air, respectively.
[5] $I_g/I_0$—where $I_g$ and $I_0$ are the electrical currents measured in the presence of gas and synthetic air, respectively. [6] $(R_g-R_0)/R_0 \cdot 100\%$.

**Table A4.** Comparison between three nanocomposites.

| Material Properties | $SrTiO_3$ | $BaTiO_3$ | $BaSrTiO_3$ |
|---|---|---|---|
| Type of structure | simple cubic perovskite structure [24] | cubic perovskite-type structure [46] | perovskite structure [65,66] |
| Dielectric constant $\varepsilon_0$ | $\varepsilon_0 = 300$ [25,42] | dielectric constant depends on the type of synthesis, temperature, frequency and dopants [47,48] At RT $\varepsilon_0 = 2570$ [134] | $\varepsilon_0 = 420$ [135] |
| Dielectric loss | mostly < 0.02 [25] | 0.003 [134] | 0.017 [135] |
| Application of the material | - sensors<br>- actuators<br>- electro-optical devices<br>- memory devices with random access<br>- multilayer capacitors [27]<br>- oxygen sensors [28]<br>- temperature sensors [29]<br>- cantilever base for various sensors [30] | - ferroelectric memories [56]<br>- electro-optical devices [57]<br>- dielectric capacitors [58]<br>- multilayer capacitors (MLCs) [59]<br>- electromechanical transducers [60,61]<br>- gas sensor applications [62] | - electronic components<br>- ferroelectric memories<br>- capacitors<br>- phase shifters [65,66]<br>- gas-sensitive material [67]<br>- multilayer and voltage-tunable capacitors<br>- dynamic random access memories (DRAM) [136] |
| Deposition method | - magnetron sputtering [32,33]<br>- atomic layer deposition (ALD) [34,35]<br>- pulsed laser deposition (PLD) [36,37]<br>- metal-organic chemical vapor deposition (MOCVD) [38,39]<br>- laser chemical vapor deposition (LCVD) [40,41],<br>- sol–gel method [42,43] | - solid-state reaction [50]<br>- sol–gel method [51]<br>- hydrothermal method [52]<br>- coprecipitation method [53]<br>- polymeric precursor method [54]<br>- mechanochemical synthesis [55] | - spin coating [67]<br>- pulsed laser deposition (PLD) [70]<br>- chemical solution deposition (CSD) [71]<br>- RF magnetron co-sputtering process [99]<br>- RF sputtering technique [94]<br>- hydrothermal method [106]<br>- screen printing technique [104]<br>- mechanochemical process [132] |
| Sensitivity effect to | propane [76]<br>propen [76]<br>NO [76]<br>$O_2$ [77]<br>$O_3$ [78]<br>CO [79]<br>$H_2S$ [80]<br>$CO_2$ [81]<br>volatile organic compounds (VOC), such as ethanol [82] | CO [90]<br>$CO_2$ [94]<br>$NO_2$ [95]<br>$NH_3$ [95]<br>LPG [96]<br>$H_2$ [96]<br>$H_2S$ [96,97]<br>Humidity [98]<br>$NO_x$ [123] | $NH_3$ [68,69,101,102]<br>$H_2S$ [68,70,100]<br>$H_2$ [99]<br>Humidity [104–106]<br>$NO_2$ [100,107]<br>LPG [107] |
| Other advantages | low cost and strong stability in thermal and chemical atmospheres [26,42]<br>fiber optic evanescent-wave hydrogen sensors [83]<br>operating solid oxide fuel cell (SOFC) [83]<br>microwave-based gas sensors [84] | large electro-optic coefficients, positive coefficient of resistivity (PTCR) [49] | high thermal and chemical stability [100], good tenability tunable filters, oscillators microwave phase shifters uncooled infrared sensors, etc. [136] |

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
