# Peer review of "Semiconducting Metal Oxides: SrTiO3, BaTiO3 and BaSrTiO3 in Gas-Sensing Applications: A Review"

_coatings, doi:10.3390/coatings11020185_

Round 1

Reviewer 1 Report

Dear authors,

your paper intitled "Semiconducting metal oxides: SrTiO3, BaTiO3, and BaSrTiO3 in gas-sensing applications: a review." is very interesting and well-written.

To my side there are only some remarks to provide during minor revisions stage:
- the last tables: please include the thickness and the area of the powder /sensor layer or at least provide "thick film" or "thin film"; I think that for screen-printing is difficult to obtain "thin film" less than 1 micrometer, also providing a solution to screen with solvent;
- decrease the dimensions of the Figure 2.

Best regards.

Author Response

Dear Reviewer,

we wish to express great thanks for your fruitful comments on our paper. We appreciate very much your relevant comments and suggestions.

Reviewer 2 Report

This manuscript presents an overview of the state-of-the-art recent research advances on SrTiO3, BaTiO3, and BaSrTiO3 nanoparticles for gas sensing applications. They reviewed the various available techniques of gas sensors with these nanoparticles. This is a timely, interesting and important review article. It has been well prepared and organized. I recommend accept after minor revisions.

  1. In the introduction, I would recommend to reorganize the order of sentences, because several topics are placed back and forth, so it is somewhat hard to follow the story flow.
  2. I suggest the authors provide a paragraph about the advantages of these materials than others. The comparison between these three nanoparticles-based gas sensors should be described.
  3. Authors show many kinds of references and well introduce about the various techniques around this field. Although, some relevant papers on issue of gas sensing should be included like- Sustainable Materials and Technologies, 2018, 16, 1-11. Carbon, 2016, 110, 97-129.

4.The part of Summary and Perspectives should be increases.

  1. Images quality should be improved.

Author Response

(The authors gave the same response as above.)

Reviewer 3 Report

This is a focused review on specific gas sensing materials. It is a good summary. 

I would like to see one figure where the basic mechanisms are presented. Which material property of those materials makes then specifically unique. there is range of related niobate and titanate materials which are widely used due to their piezo-electric properties. optical sensors are made using them.  the scope of materials limiting only to the selected three could be more explained. 

in the tables at the end, please, define sensitivity units.  

Author Response

(The authors gave the same response as above.)
